# Cellular and Humoral Responses to Recombinant and Inactivated SARS-CoV-2 Vaccines in CKD Patients: An Observational Study

**DOI:** 10.3390/jcm12031225

**Published:** 2023-02-03

**Authors:** Siliang Zhang, Jiaoxia He, Bin Tang, Qin Zhou, Yudong Hu, Yuan Yu, Jianwei Chen, Yi Liu, Chunmeng Li, Hong Ren, Xiaohui Liao

**Affiliations:** 1Department of Nephrology, Second Affiliated Hospital, Chongqing Medical University, Chongqing 400010, China; 2Department of Infectious Diseases, Key Laboratory of Molecular Biology for Infectious Diseases (Ministry of Education), Institute for Viral Hepatitis, The Second Affiliated Hospital, Chongqing Medical University, Chongqing 400010, China

**Keywords:** SARS-CoV-2 vaccine, safety, cellular response, CKD, humoral responses

## Abstract

Background: It remains unclear what B cell and humoral responses are mounted by chronic kidney disease (CKD) patients in response to recombinant and inactivated SARS-CoV-2 vaccines. In this study, we aimed to explore the cellular and humoral responses, and the safety of recombinant and inactivated SARS-CoV-2 vaccines in CKD patients. Methods: 79 CKD and 420 non-CKD individuals, who completed a full course of vaccination, were enrolled in the study. Adverse events (AEs) were collected via a questionnaire. Cellular and humoral responses were detected at 1, 3, and 6 months, including IgG antibody against the receptor-binding domain (RBD) of the SARS-CoV-2 spike protein (anti-RBD-IgG), neutralizing antibodies (NAbs), the positive rate of NAbs and anti-RBD-IgG, RBD-atypical memory B cells (MBCs) (CD3 − CD19 + RBD + CD21 − CD27−), RBD-activated MBCs (CD3 − CD19 + RBD + CD21 − CD27+), RBD-resting MBCs (CD3 − CD19 + RBD + CD21 + CD27+), and RBD-intermediate MBCs (CD3 − CD19 + RBD + CD21 + CD27−). Results: We found no differences in the positivity rates of NAbs (70.89% vs. 79.49%, *p* = 0.212) and anti-RBD IgG (72.15% vs. 83.33%, *p* = 0.092) between the CKD and control groups. A total of 22 CKD individuals completed the full follow-up (1, 3, and 6 months). Significant and sustained declines were found at 3 months in anti-RBD IgG (26.64 BAU/mL vs. 9.08 BAU/mL, *p* < 0.001) and NAbs (161.60 IU/mL vs. 68.45 IU/mL *p* < 0.001), and at 6 months in anti-RBD IgG (9.08 BAU/mL vs. 5.40 BAU/mL, *p* = 0.064) and NAbs (68.45 IU/mL vs. 51.03 IU/mL, *p* = 0.001). Significant differences were identified in MBC subgroups between CKD patients and healthy controls, including RBD-specific atypical MBCs (60.5% vs. 17.9%, *p* < 0.001), RBD-specific activated MBCs (36.3% vs. 14.8%, *p* < 0.001), RBD-specific intermediate MBCs (1.24% vs. 42.6%, *p* < 0.001), and resting MBCs (1.34% vs. 22.4%, *p* < 0.001). Most AEs in CKD patients were mild (grade 1 and 2) and self-limiting. One patient with CKD presented with a recurrence of nephrotic syndrome after vaccination. Conclusions: The recombinant and inactivated SARS-CoV-2 vaccine was well-tolerated and showed a good response in the CKD cohort. Our study also revealed differences in MBC subtypes after SARS-CoV-2 vaccination between CKD patients and healthy controls.

## 1. Introduction

Coronavirus disease 2019 (COVID-19) caused by the severe acute respiratory syndrome coronavirus 2 (SARS-CoV-2) is an infectious disease that has extensively impacted human health worldwide. Several studies have reported that chronic kidney disease (CKD) is a significant risk factor for hospital admission and mortality following infection with COVID-19 [1,2]. Among those infected with SARS-CoV-2, patients undergoing dialysis, with a history of organ transplantation, and with an estimated glomerular filtration rate (eGFR) < 30 mL/min/1.73 m^2^ appear to have higher likelihood for worse outcomes [3]. Immunological dysfunction, which is a feature of CKD that is exacerbated by decreased eGFR, is likely of multi-factorial origins including chronic inflammation, endothelial cell dysfunction, uremia, malnutrition, and cytokine deregulation. Vaccination is a critical component of the defense against infection. Individuals with CKD benefit from vaccinations against hepatitis B, influenza, pneumococcal disease, and herpes zoster [4]. Several studies have demonstrated good responses to mRNA vaccines in non-dialysis dependent patients, hemodialysis and peritoneal dialysis patients. Current clinical evidence also supports that mRNA vaccines are safe in CKD patients [5,6].

There are areas of potential concern, however. Memory B cells (MBCs), which maintain long-lasting immunity, are an important component of the humoral and cellular response to SARS-CoV-2 [7]. However, few studies have tested the MBCs response to the SARS-CoV-2 vaccine [8]. This makes it difficult to evaluate the duration of protective immunity resulting from the vaccination. Thus, further research is necessary to explain if and whether these cell populations affect the protective responses and incidence of adverse reactions following vaccination.

Inactivated and recombinant SARS-CoV-2 vaccines have been widely used in many countries around the world, including China. However, few observational studies have focused on the safety and efficacy of these vaccines in CKD populations [9]. The MBCs response characteristics of CKD patients to SARS-CoV-2 vaccination also remain unknown. In this observational study, we report on the antibody levels and MBCs’ responses to SARS-CoV-2 inactivated and recombinant vaccines in CKD patients with and without hemodialysis.

## 2. Methods

### 2.1. Participants

Healthy individuals, those with non-dialysis-dependent CKD, and those undergoing hemodialysis were recruited into this observational study between 1 August 2021 and 31 December 2021 from the Second Affiliated Hospital of Chongqing Medical University, China. CKD participants had to meet the clinical diagnostic criteria of the KDIGO guidelines for CKD. According to the preliminary results, the positive rate of antibody in the non-CKD group and the CKD group was 86% and 65%, respectively. With 10% of the loss of follow-up involved in calculating, more than 75 cases were required for each the non-CKD group and the CKD group. Recombinant and inactivated vaccine recipients who received the SARS-CoV-2 vaccine within 3 months were enrolled; vaccine subtypes were used as an independent variable. The following were exclusionary: (1) history of COVID-19 or positive SARS-CoV-2 nucleic acid amplification test; (2) close contact with SARS-CoV-2 infected individuals; (3) current pregnancy; (4) did not complete the full-course of vaccination. The study was approved by the Ethics Committee of the Second Affiliated Hospital of Chongqing Medical University and conformed to the ethical guidelines of the Declaration of Helsinki (Ratification No. 111/2021). All participants provided written informed consent before participation. The study was registered at ClinicalTrials.gov NCT05043246 and the follow ups are ongoing.

### 2.2. Data Collection

Electronic questionnaires and e-cases were used to obtain patient demographic, adverse events, and clinical data. The questionnaire of adverse events is shown in Appendix A. Information about the patient’s gender, age, time of vaccination to sample collection, body mass index, type of vaccine, comorbidities, etc. were collected. Time intervals after the full course of vaccination were defined as 1 month (=21–45 days), 3 months (=76–105 days), and 6 months (=165–195 days).

### 2.3. SARS-CoV-2 Antibody Test

Plasma samples were collected to detect IgG antibodies against the receptor-binding domain (RBD) of the SARS-CoV-2 spike protein (anti-RBD-IgG) and neutralizing antibodies (NAbs) using capture chemiluminescence immunoassays (MAGLUMI X8, Snibe, Shenzhen, China) according to the manufacturer’s instructions. The manufacturer of the kit (130219017M, 130619017M, MAGLUMI X8, Snibe, Shenzhen, China) reported that the anti-S-RBD-IgG tests have a 100% sensitivity and 99.6% specificity for the diagnosis of COVID-19, while the manufacturer of the kit (130219027M, 130619027M,MAGLUMI X8, Snibe, Shenzhen, China) reported that NAbs tests have a 100% sensitivity and 100% specificity. The cut-off values were 4.33 BAU/mL for Anti-RBD-IgG and 60.75 IU/mL for NAbs. The detailed procedures used to process each kit are shown in Appendix A.

### 2.4. Detection of SARS-CoV-2 Specific B Cells by Flow Cytometry

For SARS-CoV-2 specific B cell detection, biotinylated SARS-CoV-2 Spike RBD protein (40592-V08H2-B, Sino Biological, Beijing, China) was mixed with Streptavidin-BV421 (405225, Biolegend, San Diego, CA, USA) at a 4:1 molar ratio for one hour at 4 °C to obtain the antigen probe. According to the manufacturer’s instructions, peripheral blood mononuclear cells (PBMCs) were isolated from heparinized (sodium heparin) fresh whole blood (the time from arm to cell isolation was <4 h) by Histopaque (10771, Sigma-Aldrich, MI, USA) density gradient centrifugation. After washing with FACS buffer (PBS+2% FBS [FSD500, Excell Bio, Shanghai, China]), approximately 0.5 × 10^6^ PBMCs were incubated with the antigen probe (biotinylated SARS-CoV-2 Spike RBD protein-Streptavidin-BV421) or incubated with Streptavidin-BV421 alone as a negative control, for 30 min at 4 °C and the following conjugated antibodies: PerCP/Cyanine5.5 conjugated anti-human CD3 (1:50, 300430, Biolegend, San Diego, CA, USA), APC conjugated anti-human CD19 (1:50, 302212, Biolegend), Alexa Fluor^®^ 700 conjugated anti-human CD21 (1:50, 354918, Biolegend), and PE conjugated anti-human CD27 (1:50, 356406, Biolegend). After staining, cells were washed and resuspended in 200μL of FACS buffer. Sample data were then acquired by flow cytometry (Beckman Coulter, CytoFLEX, Brea, CA, USA) and analyzed using FlowJo (Treestar, 10.0.7r2). Lymphocytes were sorted by utilizing FSC and SSC gated channels. The gating strategy for RBD-specific B cells was based on the negative control. RBD-specific memory B cells (MBCs) were divided into four subsets based on the expression of CD27 and CD21. The cell populations were identified by using the following strategy: RBD-specific B cells (CD3 − CD19 + RBD+), RBD-specific MBCs (CD3 − CD19 + RBD + CD27+), RBD-atypical MBCs (CD3 − CD19 + RBD + CD21 − CD27−), RBD-activated MBCs (CD3 − CD19 + RBD + CD21 − CD27+), RBD-resting MBCs (CD3 − CD19 + RBD + CD21 + CD27+), and RBD-intermediate MBCs (CD3 − CD19 + RBD + CD21 + CD27−). The full gating strategy for the target cell populations is shown in Appendix A. Examples of flow plots are presented in Appendix A.

### 2.5. Statistical Analysis

Antibody and MBCs levels were compared between the CKD and healthy control groups. As a subgroup analysis, we contrasted the responses of CKD patients receiving inactivated vaccine and dialysis-dependent CKD patients versus healthy controls. Due to differences in baseline and sample sizes between the CKD and control groups, 1:1 Propensity Score Matching (PSM) was utilized to screen and reduce potential bias arising from baseline differences [10]. Categorical variables were analyzed by using Chi-square or Fisher’s precision probability tests. Independent samples *t*-tests were used to compare normally distributed continuous variables, while the Mann–Whitney test was used for non-normally distributed data. A *p*-value < 0.05 was considered statistically significant. Data analysis was performed by using IBM SPSS 25.0 (Armonk, NY, USA). Data were visualized using Graphpad 7.0.

## 3. Results

### 3.1. Characteristics of Participants

We recruited 79 CKD individuals, including 22 with dialysis-dependent CKD and 57 with non-dialysis-dependent CKD. 420 non-CKD individuals were enrolled as controls. In the baseline comparison, the proportions of gender and vaccine type were found to differ between the CKD and control groups and thus 1:1 PSM was used. The prevalence of diabetes, hypertension, and cardiovascular disease was higher in the CKD group. However, diabetes, hypertension, and cardiovascular disease were not included in the PSM model, as the prevalence was too low in the controls. Subgroup analyses of the inactivated vaccine group and the hemodialysis CKD patient group were conducted. The baseline characteristics for the overall CKD patient and healthy control groups are shown in Table 1. The baseline characteristics for the inactivated vaccine and healthy control groups are shown in Table 2. The hemodialysis and healthy control group characteristics are shown in Table 3. Due to China’s vaccination policy against COVID-19, we were only able to collect 17 healthy unvaccinated cases from the community. We compared NAbs levels between the unvaccinated and vaccinated groups. These data are shown in Appendix A.

### 3.2. SARS-CoV-2 Vaccination in CKD Patients

Using 1:1 PSM, 79 healthy individuals were included in the healthy control group. Humoral immunity was assessed by comparing Anti-RBD IgG and NAbs in the CKD and control groups. The positivity rates of NAbs (70.89% vs. 79.49%, *p* = 0.212) and anti-RBD IgG (72.15% vs. 83.33%, *p* = 0.092) were found to not differ between groups (Figure 1B,D). Additionally, there was no significant difference in anti-RBD IgG levels between the CKD and the control groups (13.70 BAU/mL vs. 18.96 BAU/mL, *p* = 0.089), while NAbs levels were lower in the CKD group (96.39 IU/mL vs. 127.58 IU/mL, *p* = 0.046) (Figure 1A,C).

MBCs are recognized as a crucial component of cellular and humoral responses in virological immunity. We next assessed the frequency of RBD-specific MBCs in each group. There were no differences in RBD-specific MBCs levels between the CKD and control groups (Appendix A). MBCs were divided into four subsets according to CD21 and CD27 expression [11]. Intermediate MBCs and resting MBCs with the non-plasmablast population express CD21+; there are two related subsets, including a CD21− CD27+ population with plasmablast-like features, and a CD21− CD27− population, which are deemed to be atypical memory B cells. The frequencies of RBD-specific atypical MBCs (60.5% vs. 17.9%, *p* < 0.001) and RBD-specific activated MBCs (36.3% vs. 14.8%, *p* < 0.001) were higher in the CKD group (Figure 1E,G). The frequencies of RBD-specific intermediate MBCs (1.24% vs. 42.6%, *p* < 0.001), and resting MBCs (1.34% vs. 22.4%, *p* < 0.001) were higher in the control group (Figure 1F,H).

22 individuals with CKD completed the full follow-up from 1 month to 6 months after vaccination. We found a sustained reduction between 1 and 3 months for anti-RBD IgG (26.64 BAU/mL vs. 9.08 BAU/mL, *p* < 0.001), NAbs (161.60 IU/mL vs. 68.45 IU/mL, *p* < 0.001), and positive rate of NAbs (95.45% vs. 63.64%, *p* = 0.021) and anti-RBD IgG (77.27% vs. 31.82%, *p* = 0.006). We also found a reduction between 3 and 6 months for anti-RBD IgG (9.08 BAU/mL vs. 5.40 BAU/mL, *p* = 0.064) and Nabs (68.45 IU/mL vs. 51.03 IU/mL, *p* = 0.001) (Figure 2).

### 3.3. Inactivated SARS-CoV-2 Virus Vaccination in CKD

Using 1:1 PSM, 63 healthy individuals were included as healthy controls. In the subgroup that received the inactivated vaccine, we found no significant differences in anti-RBD IgG (12.61 BAU/mL vs. 11.49 BAU/mL, *p* = 0.469) or NAbs (88.29 IU/mL vs. 98.42 IU/mL, *p* = 0.188) levels in the CKD versus the control group (Figure 3A,C). The positivity rates of NAbs (60.32% vs. 78.33%, *p* = 0.0634) and anti-RBD IgG (68.25% vs. 79.03%, *p* = 0.1717) were not different between groups (Figure 3B,D). The CKD group showed higher levels of RBD-specific atypical MBCs (57.75% vs. 18%, *p* < 0.001) and RBD-specific activated MBCs (38.05% vs. 15.2%, *p* < 0.001) than controls (Figure 3E,G), and showed lower RBD-specific intermediate MBCs (1.28% vs. 40.65%, *p* < 0.001) and resting MBCs (1.20% vs. 24.2%, *p* < 0.001) than controls (Figure 3F,H).

### 3.4. SARS-CoV-2 Vaccination in Hemodialysis Patients

Hemodialysis patients were expected to display a lower vaccine response. Due to the heterogeneity of the included groups, 1:1 PSM might have excessively reduced the sample size. Therefore, in this subgroup analysis, we did not perform sample matching. The positivity rate of anti-RBD IgG (69.57% vs. 89.1%, *p* = 0.0132) was lower in hemodialysis patients, while the positivity rate of NAbs (69.57% vs. 79.89%, *p* = 0.2854) was not significantly different between hemodialysis patients and controls (Figure 4B,D). The levels of anti-RBD IgG (16.75 BAU/mL vs. 24.51 BAU/mL, *p* = 0.011) were lower in the hemodialysis group, while the levels of NAbs (95.18 IU/mL vs. 130.21 IU/mL, *p* = 0.061) were not different between groups (Figure 4A,C). The CKD group showed higher levels of RBD-specific atypical MBCs (61.7% vs. 19.5%, *p* < 0.001) and RBD-specific activated MBCs (33.9% vs. 15.7%, *p* < 0.001) than controls (Figure 4E,G), and showed fewer RBD-specific intermediate MBCs (1.89% vs. 41%, *p* < 0.001) and resting MBCs (1.07% vs. 21.4%, *p* < 0.001) than controls (Figure 4F,H).

### 3.5. Safety of SARS-CoV-2 Vaccination in CKD

There was a difference in the overall incidence of AEs between the CKD and healthy control groups (6.3% vs. 13.1%, *p* = 0.09) (Table 4). Most recorded AEs in the CKD group were mild (grades 1 and 2) and self-limiting; these included fatigue, slight fever, and nausea. Proteinuria occurred in two patients with CKD, both of whom had preexisting chronic glomerulonephritis. One of the patients progressed to nephrotic syndrome and required immunosuppressive therapy. In the healthy control group, we did not find any moderate or severe AEs (grades 3 and 4).

## 4. Discussion

In this study, anti-RBD-IgG and NAbs were used to comprehensively analyze the humoral immune response to SARS-CoV-2 vaccination. We report that the positive rate of NAbs and anti-RBD IgG were 70.89% and 72.15%, respectively, in CKD patients following two doses of the vaccine. In our subgroup analysis for the CKD population using inactivated vaccines, we achieved a similar result. In the hemodialysis subgroup, the positive rate of anti-RBD IgG was significantly lower than in healthy controls. In additional analyses of antibody levels, NAbs levels were lower in the CKD group and hemodialysis subgroup, and anti-RBD IgG levels were lower in the CKD group. These results are consistent with studies assessing responses to COVID19 vaccines. In data from Israel’s largest healthcare organization, a 74% protection rate against the subsequent development of severe disease was reported after two doses of the BNT162b2 mRNA vaccine [12]. Chung et al. reported that 94.16% of maintenance dialysis patients without prior SARS-CoV-2 infection achieved a positive antibody response after two doses of the ChAdOx1 nCoV-19 Vaccine [13]. The RECOVAC immune-response study reported a high seroconversion rate in participants with CKD G4/5 (100%) and dialysis (99.4%), which was similar to controls. A factor analysis of several studies showed that older age and immunosuppressive treatment were risk factors for reduced vaccine response rates [6].

In the follow-up cohort, we found a rapid decrease in antibody levels over time. Quiroga et al. reported a sustained decline of anti-spike antibody titers at 1, 3 and 6 months in CKD patients following two doses of the BNT162b2 mRNA vaccine [14]. Zhang et al. reported a longitudinal analysis of T cell, B cell, and antibody responses to four different SARS-CoV-2 vaccines in humans and concluded that mRNA vaccines were associated with substantially reduced antibodies at 6 months, but memory T cell and B cell levels remained fairly stable [15]. Our study shows a low immune response to vaccines in the CKD population. Nevertheless, it does not dismiss the importance of vaccination in protecting CKD individuals, who are more vulnerable to COVID-19 sequelae. 

MBCs are antigen exposed cells with the ability to generate a more rapid and effective immune response during secondary antigen exposure. RBD-specific MBCs were not found to be significantly different between CKD and healthy controls. As a subset of MBCs, CD21 + CD27 + MBCs play a central role in humoral immune responses and can rapidly differentiate into antibody-secreting plasma cells. CD21 + CD27− intermediate MBCs, a naïve subset with RBD-specific surface Ig receptors, can be activated following ligation of their cognate antigens through the antigen-specific B cell receptors (BCRs), and enter into germinal centers to undergo affinity maturation [16,17]. We found significantly lower CD21+ CD27+ MBCs and CD21+ CD27− MBCs in CKD patients. We are still uncertain how such changes impact the immune memory of cells in CKD patients. Two poorly understood subsets of MBCs are CD21− CD27+ activated MBCs or the plasmablast-like subset and the CD21− CD27− population or atypical MBCs [16,17]. We found an expansion of atypical CD21− CD27− MBCs and activated CD21− CD27+ MBCs in CKD compared to the healthy control group following vaccination. This shows a discrepancy in cellular immunological mechanisms between CKD patients and healthy controls. Currently, the function of atypical MBCs remains unclear. Atypical MBCs, considered a subset of MBCs, are usually seen at high frequencies in chronic diseases [18]. Several studies have reported remarkable increases of CD21− CD27− atypical B cells in chronic infectious diseases, such as malaria, HIV-AIDS, tuberculosis (TB), and several autoimmune conditions [19,20]. A previous study showed that atypical memory B cells are short-lived activated cells that may represent a precursor plasma cell (PC) population [18]. 

Several studies into rheumatic diseases have reported that atypical MBCs may be a pathogenic immune factor of disease-causing antibodies. Chunmei et al. reported that greater numbers of atypical MBCs were associated with high disease flares and disease-specific autoantibodies such as anti-Smith (Sm) antibodies. Atypical MBCs are also found to infiltrate kidneys in lupus nephritis and to be closely associated with disease activity and renal dysfunction [21]. Cloé et al. demonstrated that TLR9 signaling in HCV-associated atypical memory B cells triggers Th1 and rheumatoid factor autoantibody responses. It appears that ongoing chronic inflammation promotes the generation of these MBCs [22]. However, in infectious diseases, evidence suggests that atypical memory B cells are positive alternatives that participate in mounting defenses against pathogens. Christine S et al. reported that in acute febrile malaria, specific atypical MBCs and activated MBCs up-regulate similar intracellular signaling cascades to stimulate differentiation into antibody-secreting cells and the up-regulation of molecules that mediate B-T cell interactions. With the T follicular helper cells and staphylococcal enterotoxin B, atypical MBCs can differentiate into CD38+ antibody-secreting cells in vitro [23]. Similar supporting evidence has been found in single-cell sequencing studies; that atypical B cells are part of a broader alternative lineage that is abundant even in healthy individuals, and that they are a critical component of the humoral immune response [16]. Kathryn A et al. reported that SARS-CoV-2 infection results in the production of more atypical MBCs than when the immune system is primed by mRNA vaccines [24]. Atypical memory B cells appear to be an important cell subset for developing humoral immunity in response to the SARS-CoV-2 vaccine. More research is needed to explore their pros and cons.

Most of our CKD patients showed a good tolerance after vaccination. Most adverse events were mild and self-limiting. Importantly, in the CKD group, two individuals presented with recurrent proteinuria, one of whom progressed to nephrotic syndrome and required immunosuppressive therapy. A small number of case reports have reported de novo or recurrent glomerulonephritis following SARS-CoV-2 vaccination; the proteinuria gradually improved without any medication, suggesting that immune activation by the vaccine is unlikely to elicit a marked progression of glomerulonephritis.

Our study is the first in which healthy people were utilized as controls and focused on the cellular and humoral responses to inactivated and recombinant SARS-CoV-2 vaccination in CKD patients. These data are important for clinical risk-benefit decision-making. We also evaluated the subtypes of MBC responses to explore the underlying antibody responses. There were some limitations. First, only hemodialysis and non-dialysis dependent patients were included; renal transplant recipients and peritoneal dialysis patients were not enrolled. Second, it was a low sample size study, and only 22 patients completed the 6-month antibody test follow-up. Due to the lack of follow-up data for healthy controls, only the CKD group levels were available for comparison. Third, although baseline matching was performed by 1:1 PSM, the prevalence of cardiovascular disease, hypertension, and diabetes were significantly higher in the CKD group than in the healthy control group, which may have introduced bias.

In conclusion, we analyzed the antibody response, B cell response, and safety profile of recombinant and inactivated SARS-CoV-2 vaccines in patients with CKD and healthy controls. After completing a full vaccination course, we found that the recombinant and inactivated anti-SARS-CoV-2 vaccines were well tolerated and showed good responses in majority of the CKD population. Nevertheless, we found a decrease in antibody levels at 3 months post-vaccination. We also found differences in MBCs subtypes after SARS-CoV-2 vaccination between CKD patients and healthy controls. The differences in subgroups of MBCs between CKD patients and healthy individuals deserves further study. Based on our findings, we believe that it is essential to develop a vaccination strategy that is appropriate for people with CKD. Additionally, the B-cell signatures of CKD patients will be an important inspiration for revealing the pathological immunological mechanisms of the CKD population. 

## Figures and Tables

**Figure 1 jcm-12-01225-f001:**
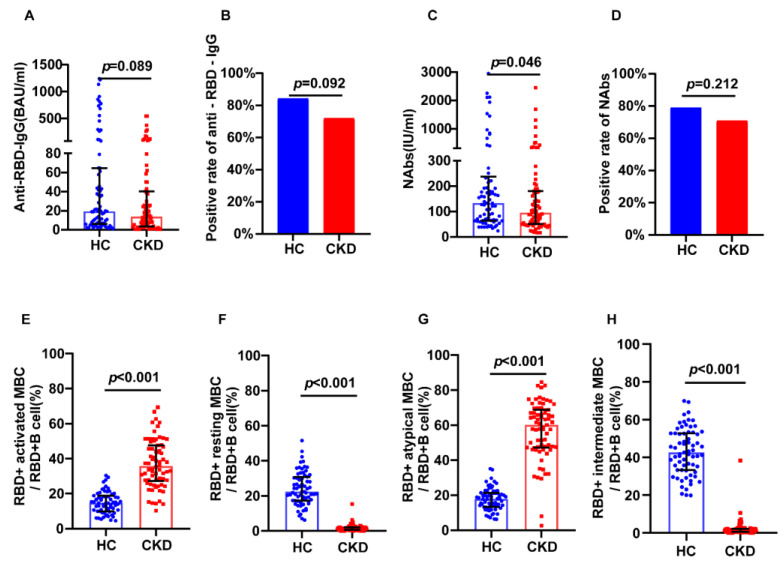
Humoral immune responses following vaccination in CKD patients and healthy controls. (**A**) The serum anti-RBD-IgG levels. (**B**) The seropositivity rates of anti-RBD-IgG. (**C**) The Serum NAbs levels. (**D**) The seropositivity rates of Nabs. (**E**) The frequency (percentage of RBD-specific B cells) of RBD-specific activated MBCs. (**F**) The frequency (percentage of RBD-specific B cells) of RBD-specific resting MBCs. (**G**) The frequency (percentage of RBD-specific B cells) of RBD-specific atypical MBCs. (**H**) The frequency (percentage of RBD-specific B cells) of RBD-specific intermediate MBCs responses. The IQR are indicated by error bars. anti-RBD-IgG, spike receptor-binding domain IgG antibody; CKD chronic renal disease; HC healthy controls; IQR interquartile range; MBCs memory B cells; NAbs neutralizing antibodies.

**Figure 2 jcm-12-01225-f002:**
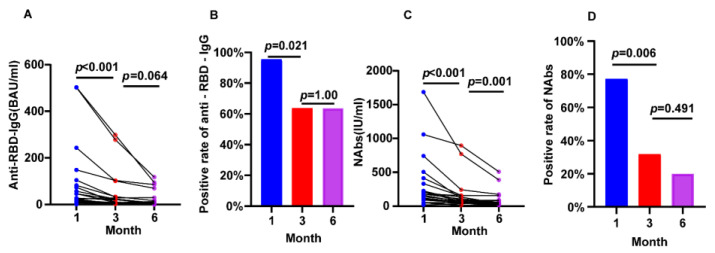
Longitudinal changes of humoral immune responses following immunization with vaccines in CKD patients. (**A**) The change of serum anti-RBD-IgG levels. (**B**) The change of seropositivity rates of anti-RBD-IgG. (**C**) The change of serum NAbs levels. (**D**) The change of seropositivity rates of NAbs. The IQR are indicated by error bars. anti-RBD-IgG, spike receptor-binding domain IgG antibody; CKD chronic renal disease; NAbs neutralizing antibodies.

**Figure 3 jcm-12-01225-f003:**
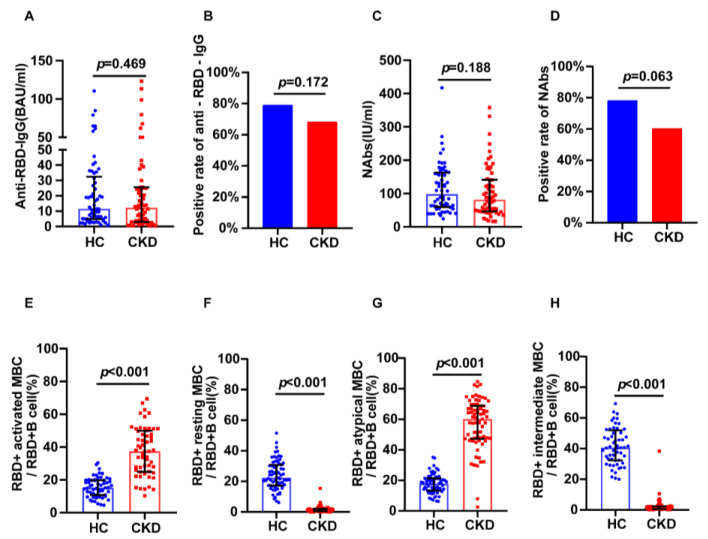
Humoral immune responses following immunization with inactivated vaccines in CKD patients and healthy controls. (**A**) The serum anti-RBD-IgG levels. (**B**) The seropositivity rates of anti-RBD-IgG. (**C**) The serum NAbs levels. (**D**) The seropositivity rates of Nabs. (**E**) The frequency (percentage of RBD-specific B cells) of RBD-specific activated MBCs. (**F**) The frequency (percentage of RBD-specific B cells) of RBD-specific resting MBCs. (**G**) The frequency (percentage of RBD-specific B cells) of RBD-specific atypical MBCs. (**H**) The frequency (percentage of RBD-specific B cells) of RBD-specific intermediate MBCs. The IQR are indicated by error bars. anti-RBD-IgG, spike receptor-binding domain IgG antibody; CKD chronic renal disease; HC healthy controls; IQR interquartile range; MBCs memory B cells; NAbs neutralizing antibodies.

**Figure 4 jcm-12-01225-f004:**
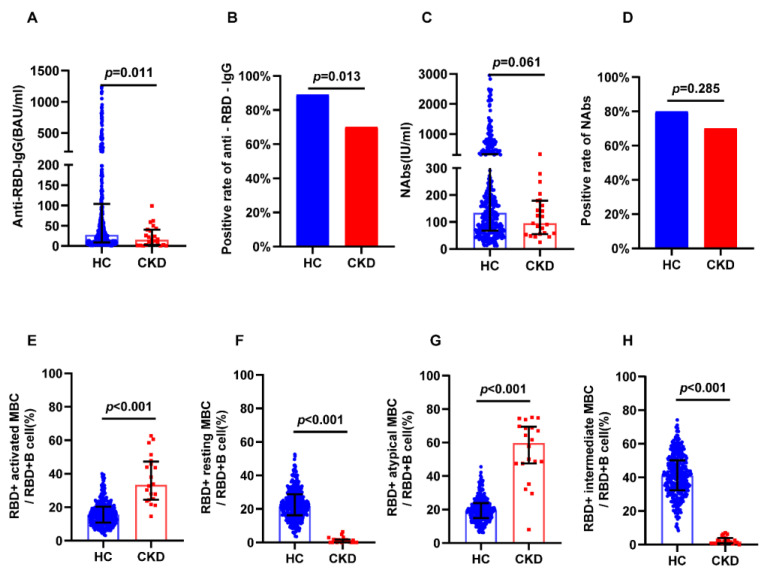
Humoral immune responses following immunization with vaccines in dialysis CKD patients and all healthy controls. (**A**) The serum anti-RBD-IgG levels. (**B**) The seropositivity rates of anti-RBD-IgG. (**C**) The serum NAbs levels. (**D**) The seropositivity rates of Nabs. (**E**) The frequency (percentage of RBD-specific B cells) of RBD-specific activated MBCs. (**F**) The frequency (percentage of RBD-specific B cells) of RBD-specific resting MBCs. (**G**) The frequency (percentage of RBD-specific B cells) of RBD-specific atypical MBCs. (**H**) The frequency (percentage of RBD-specific B cells) of RBD-specific intermediate MBCs. The IQR are indicated by error bars. anti-RBD-IgG, spike receptor-binding domain IgG antibody; CKD chronic renal disease; HC healthy controls; IQR interquartile range; MBCs memory B cells; NAbs neutralizing antibodies.

**Table 1 jcm-12-01225-t001:** Baseline characteristics of the CKD patients and healthy controls.

Variables	CKD Group	Control Group	*p* Value	Control Group *	*p* Value *
Age (years)	47.25 ± 14.31	47 (34–58)	0.682	42 (25–53)	0.051
≤60	64/79	334/420	0.763	67/79	0.526
>60	15/79	86/420		12/79	
Gender (male, (*n*%))	53/79 (67%)	218/420 (52%)	0.013	47/79 (59%)	0.409
BMI (kg/m^2^)	24.70 ± 4.25	25.04 (20.55–27.33)	0.381	23.85 ± 3.79	0.102
<24	36/79	151/420	0.105	45/79	0.152
≥24	43/79	269/420		34/79	
Acquisition time (months)					
<3	61/79	330/420	0.788	59/79	0.71
≥3	18/79	90/420		20/79	
Vaccines					
Recombinant vaccine (*n*%)	16/79	177/420	<0.001	18/79	0.15
Inactivated vaccine (*n*%)	63/79	243/420		61/79	
Comorbidities					
Diabetes	11/79	15/420	<0.001	0/79	<0.001
Hypertension	35/79	36/220	<0.001	4/79	<0.001
Cardiovascular diseases	14/79	3/420	<0.001	0/79	<0.001

* Presented as value after 1:1 Propensity Score Matching. Categorical variables were analyzed by using the Chi-square or Fisher’s precision probability tests. Independent samples *t*-tests were used to compare normally distributed continuous variables, while the Mann–Whitney tests were used for non-normally distributed data. *p*-values < 0.05 were considered statistically significant. BMI, Body Mass Index.

**Table 2 jcm-12-01225-t002:** Baseline characteristics of CKD patients receiving inactivated vaccine and healthy controls.

Variables	CKD Group	Control Group	*p* Value	Control Group *	*p* Value *
Age (years)	48.60 ± 14.28	49 (34–60)	0.642	48 (36–64)	0.918
≤60	51/63	184/243	0.287	43/63	0.102
>60	12/63	59/243		20/63	
Gender (male, (*n*%))	41/63	127/243	0.068	42/63	0.814
BMI (kg/m^2^)	24.56 (21.22–27.23)	24.96 ± 3.55	0.293	25.65 ± 3.02	0.064
<24	29/63	152/243	0.017	17/63	0.026
≥24	34/63	91/243		46/63	
Acquisition time (months)					
<3	48/63	166/243	0.224	53/63	0.264
≥3	15/63	77/243		10/63	
Diabetes	11/63	8/243	<0.001	0/63	<0.001
Hypertension	35/63	24/243	<0.001	3/63	<0.001
Cardiovascular diseases	14/63	2/243	<0.001	0/63	<0.001

* Presented as value after 1:1 Propensity Score Matching. Categorical variables were analyzed by using the Chi-square or Fisher’s precision probability tests. Independent samples *t*-tests were used to compare normally distributed continuous variables, while the Mann–Whitney tests were used for non-normally distributed data. *p*-values < 0.05 were considered statistically significant. BMI, Body Mass Index.

**Table 3 jcm-12-01225-t003:** Baseline characteristics of the hemodialysis CKD patients and healthy controls.

Variables	CKD Group	Control Group	*p* Value
Age (years)	48.00 ± 14.16	47 (34–58)	0.604
≤60	19/23	334/420	1
>60	4/23	86/420	
Gender (male, (*n*%))	14/23 (61%)	218/420 (52%)	0.402
BMI (kg/m^2^)	22.16 ± 4.38	25.04 (20.55–27.33)	0.007
<24			0.008
≥24			
Acquisition time (months)			
<3	21/23	330/420	0.189
≥3	2/23	90/420	
Vaccines			
Recombinant vaccine (*n*%)	3/23	177/420	0.006
Inactivated vaccine (*n*%)	20/23	243/420	
Comorbidities			
Diabetes	1/23	15/420	0.58
Hypertension	0/23	36/220	0.243
Cardiovascular diseases	0/23	3/420	1

Categorical variables were analyzed by using the Chi-square or Fisher’s precision probability tests. Independent samples *t*-tests were used to compare normally distributed continuous variables, while the Mann–Whitney tests were used for non-normally distributed data. *p*-values < 0.05 were considered statistically significant. BMI, Body Mass Index.

**Table 4 jcm-12-01225-t004:** Adverse events after SARS-CoV-2 vaccination.

	CKD Patients (*n* = 79)	Controls (*n* = 420)	*p* Value
Overall adverse events	5	55	0.09
Local adverse events			
Pain	/	27	0.021
Redness	/	4	0.384
Rash	/	7	0.248
Systemic adverse events			
Fatigue	1	6	0.9102
Dizziness	/	3	0.451
Diarrhea	/	1	0.664
Laryngeal pain	/	/	/
Cough	/	1	0.664
Chest distress	/	/	/
Chest pain	/	/	/
Chill	/	/	/
Proteinuria	2	/	0.001
Elevated blood pressure	/	/	/
Fever	1	1	0.185
Inappetence	/	/	/
Muscle pain	/	2	0.541
Nausea	1	4	0.806
Palpitation	/	/	/
Pruitus	/	/	/
Grade 3 and 4 adverse events	1	/	0.022

Categorical variables were analyzed by using Chi-square or Fisher’s precision probability tests. *p*-values < 0.05 were considered statistically significant.

## Data Availability

Original data can be accessed in Appendix A.

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
