# Peer review of "Cellular and Humoral Responses to Recombinant and Inactivated SARS-CoV-2 Vaccines in CKD Patients: An Observational Study"

_jcm, 2023, doi:10.3390/jcm12031225_

Round 1

Reviewer 1 Report

Authors analyzed the antibody response, B cell response, and safety profile of recombinant and inactivated COVID-19 vaccines. After completing a full vaccination course, observed that the recombinant and inactivated COVID-19 vaccines were well tolerated and showed good responses in the CKD population. Nevertheless, they found a decrease in antibody levels by 3 months after vaccination. They also observed differences in MBCs subtypes after COVID-19 vaccination between CKD patients and healthy controls. The paper is generally structured. However, in my opinion the paper has some shortcomings regarding some data analysis and text. Below I have provided numerous remarks on the text, as it is often vague and long-winded.

  1. Authors may also explore clinical variables associated with Seroconversion in CKD patients.
  2. Please mention ethics committee approval number.
  3. Authors should mention sample size and type of the study.
  4. There is a need for profound improvements of typos error, language, and grammar control throughout the manuscript. For example, in the abstract.

Line no. 31-32

We found no difference in the positivity rate of NAbs (70.89% vs. 79.49%, p=0.212) and anti-RBD IgG (72.15% vs. 83.33%, p=0.092) between between the CKD and control groups should be changed to “We found no difference in the positivity rate of NAbs (70.89% vs. 79.49%, p=0.212) and anti-RBD IgG (72.15% vs. 83.33%, p=0.092) between the CKD and control groups”.

Line no. 32-33

22 CKD individuals completed the the full follow-up (1, 3, and 6 months) should be changed to “CKD individuals completed the full follow-up (1, 3, and 6 months)”

Line no. 36-40

Significant and sustained declines were found at 3 months in anti-RBD IgG (26.64 BAU/ml vs. 9.08 BAU/ml , p<0.001), NAbs (161.60 IU/ml vs. 68.45 IU/ml p<0.001), and at 6 months in anti-RBD IgG (9.08 BAU/ml vs. 5.40 BAU/ml, p=0.064); Nabs (68.45 IU/ml vs. 51.03 IU/ml,p=0.001). Significant differences were identified in MBC subgroups between CKD patients and healthy controls, including RBD-specific atypical MBCs (60.5% vs. 17.9%, p<0.001), RBD-specific activated MBCs (36.3% vs. 14.8%, p<0.001), RBD-specific intermediate MBCs (1.24% vs. 42.6%, p<0.001), and resting MBCs (1.34% vs. 22.4%, p<0.001). should be changed to “Significant and sustained declines were found at 3 months in anti-RBD IgG (26.64 BAU/ml vs. 9.08 BAU/ml, p<0.001), Nabs (161.60 IU/ml vs. 68.45 IU/ml, p<0.001) and at 6 months in anti-RBD IgG (9.08 BAU/ml vs. 5.40 BAU/ml, p=0.064); Nabs (68.45 IU/ml vs. 51.03 IU/ml, p=0.001). Significant differences were identified in MBC subgroups between CKD patients and healthy controls, including RBD-specific atypical MBCs (60.5% vs. 17.9%, p<0.001), RBD-specific activated MBCs (36.3% vs. 14.8%, p<0.001), RBD-specific intermediate MBCs (1.24% vs. 42.6%, p<0.001) and resting MBCs (1.34% vs. 22.4%, p<0.001)”.

Author Response

  1. Authors may also explore clinical variables associated with Seroconversion in CKD patients.

Response: We have explored age, BMI, hemoglobin, glomerular filtration rate, and hepatic enzymes associated with Seroconversion in CKD patients. However, we did not find a correlations worth to report.

  1. Please mention ethics committee approval number.

Response: Ethics committee approval number has been added to the manuscript. It can be found at line 98.

  1. Authors should mention sample size and type of the study.

Response: We added sample size in Participants of Method, added type of the study as an observational study in title.

  1. There is a need for profound improvements of typos error, language, and grammar control throughout the manuscript. For example, in the abstract.

Response: We reviewed the manuscript thoroughly.  And it also was revised by an English language polishing agency.  We appreciate your language guidance. 

Line no. 31-32

We found no difference in the positivity rate of NAbs (70.89% vs. 79.49%, p=0.212) and anti-RBD IgG (72.15% vs. 83.33%, p=0.092) between between the CKD and control groups should be changed to “We found no difference in the positivity rate of NAbs (70.89% vs. 79.49%, p=0.212) and anti-RBD IgG (72.15% vs. 83.33%, p=0.092) between the CKD and control groups”.

Response: We fixed the error.

Line no. 32-33

22 CKD individuals completed the the full follow-up (1, 3, and 6 months) should be changed to “CKD individuals completed the full follow-up (1, 3, and 6 months)”

Response: We fixed the error.

Line no. 36-40

Significant and sustained declines were found at 3 months in anti-RBD IgG (26.64 BAU/ml vs. 9.08 BAU/ml , p<0.001), NAbs (161.60 IU/ml vs. 68.45 IU/ml p<0.001), and at 6 months in anti-RBD IgG (9.08 BAU/ml vs. 5.40 BAU/ml, p=0.064); Nabs (68.45 IU/ml vs. 51.03 IU/ml,p=0.001). Significant differences were identified in MBC subgroups between CKD patients and healthy controls, including RBD-specific atypical MBCs (60.5% vs. 17.9%, p<0.001), RBD-specific activated MBCs (36.3% vs. 14.8%, p<0.001), RBD-specific intermediate MBCs (1.24% vs. 42.6%, p<0.001), and resting MBCs (1.34% vs. 22.4%, p<0.001). should be changed to “Significant and sustained declines were found at 3 months in anti-RBD IgG (26.64 BAU/ml vs. 9.08 BAU/ml, p<0.001), Nabs (161.60 IU/ml vs. 68.45 IU/ml, p<0.001) and at 6 months in anti-RBD IgG (9.08 BAU/ml vs. 5.40 BAU/ml, p=0.064); Nabs (68.45 IU/ml vs. 51.03 IU/ml, p=0.001). Significant differences were identified in MBC subgroups between CKD patients and healthy controls, including RBD-specific atypical MBCs (60.5% vs. 17.9%, p<0.001), RBD-specific activated MBCs (36.3% vs. 14.8%, p<0.001), RBD-specific intermediate MBCs (1.24% vs. 42.6%, p<0.001) and resting MBCs (1.34% vs. 22.4%, p<0.001)”.

Response: We revised these sentences. We use NAbs as the abbreviation for neutralizing antibodies.

Reviewer 2 Report

This work investigates the cellular and humoral responses, as well as the adverse effects, derived from vaccination against SARS-CoV-2 in chronic kidney disease patients. The study provides new results, and conclusions are supported by the results. Nevertheless, there are some issues that should be addressed before publication:

1) In the "Introduction", line 50, authors state "Several studies have reported that chronic kidney disease is a significant risk factor...". Nevertheless, no references are provided in that sentence, and a single reference [1] is provided in the next. Please reference the rest of the several studies mentioned in the sentence.

2) In the "Introduction", line 65, "However, few studies have tested the MBCs response to the COVID-19 vaccine". Similarly, no reference is provided, although the sentence states that there are a few of studies investigating the topic. Please provide the reference of the few studies mentioned in the sentence.

3) Similarly, in "Introduction", line 71-72, "However, there are very few observational studies focused on the safety and efficacy of these vaccines...". Again, no reference is provided, while the sentence states that at least some studies exist.

4) In the "Introduction" section, line 51-52 of the original manuscript, authors say "...CKD is a significant risk factor [...] following infection with COVID-19. Among those infected with COVID-19...". As very clearly stated by the authors themselves, COVID-19 is the disease, not the virus. Thus, patients are infected by SARS-CoV-2, not by COVID-19. I would suggest authors change these and review other mentions of COVID-19, and change it by "SARS-CoV-2" when referring to the infection by the virus and not the disease.

5) In the "Introduction" section, line 58, "Individuals with CKD can have previously benefitted...". "Can" should be removed.

6) In the "Methods" section, "Data collection" subsection, line 96-97, "...type of vaccine, comorbidities, etc. will be collected". Authors used the future tense in this sentence. Do they refer to the follow-up specifically, which has not been completed? If not, "were collected" should be used instead.

7) In the "Methods" section, "Data collection" subsection, line 98, "3 months" is repeated twice. In the second case it should state "6 months".

8) In the "Methods" section, line 120, a space is missing between "probe" and "(biotinylated...)".

9) In the "Methods" section, line 122, what does the "(1:33.3)" ratio refer to?

10) The questionnaire used for assessment of adverse effects is not detailed in the manuscript, nor in the Supplementary material. The questionnaire should be added to evaluate possible bias.

11) In the "Results" section, line 169, "Presents as value ..." should say "Presented as value...". The same is true for table 2, line 177.

12) In the "Results" section, line 210, "Nabs" should be "NAbs".

13) In the "Results" section, line 257-258, authors state "the positivity rate of NAbs (69.57% vs 79.89%, p=0.2854) was significantly different...". By the definition established by the authors in the "Methods" section, a p value of 0.2854 should not be significant.  Is the stated p value correct?

14) In the "Discussion" section, lines 294-295, "...but this sound caution on this result" should be corrected.

15) In the "Discussion" section, line 303, "The RECOVAC immune-responde atudy..." should be corrected to "study".

16) In the "Discussion" section, when referencing work by other groups, authors mention their first name, or first name and last name. For example, "ChungYi et al." (line 301), "Borja Quiroga et al." (line 308), "Zeli Zhang et al." (line 310), among others. Traditionally, when citing works by other authors in text, usually only the surname is used. For example "Quiroga et al.".

17) In the "Discussion" section, line 316, the comma after "MBCs" should be removed. In the same line, a space should be added between "exposed" and "cells".

18) In the "Discussion" section, line 359-361, "The CKD patients showed good tolerance... two individuals presented with recurrent proteinuria, one of whom progressed to nephrotic syndrome". Taking into account that two of the CKD patients presented grade 3 or 4 adverse effects, stating that "The CKD patients showed good tolerance" is problematic, as it implies that all of them showed good tolerance. Alternatively, I propose changing it to "Most of the CKD patients showed good tolerance". Similary, in line 380-381, "...vaccines were well tolerated and showed good responses in the CKD population" should be changed. For example: "...showed good responses in the majority of the CKD population".

19) Reference 5 is almost certainly incorrect. The title of the document is given as "Since January 2020 Elsevier Has Created a COVID-19 Resource Centre...", which looks like a generic message from Elsevier, probably automatically added to the first page of the referenced paper, and automatically but incorrectly detected as the title by a reference manager program. Authors should check that the references are correctly formatted.

Author Response

This work investigates the cellular and humoral responses, as well as the adverse effects, derived from vaccination against SARS-CoV-2 in chronic kidney disease patients. The study provides new results, and conclusions are supported by the results. Nevertheless, there are some issues that should be addressed before publication:

Response: We are very appreciating for your careful guidance on the language. And this manuscript also was revised by an English language polishing agency.

1) In the "Introduction", line 50, authors state "Several studies have reported that chronic kidney disease is a significant risk factor...". Nevertheless, no references are provided in that sentence, and a single reference [1] is provided in the next. Please reference the rest of the several studies mentioned in the sentence.

Response: Relevant references have been added. References 1-2 cover this.

2) In the "Introduction", line 65, "However, few studies have tested the MBCs response to the COVID-19 vaccine". Similarly, no reference is provided, although the sentence states that there are a few of studies investigating the topic. Please provide the reference of the few studies mentioned in the sentence.

Response: Relevant references have been added. References 8 cover this.

3) Similarly, in "Introduction", line 71-72, "However, there are very few observational studies focused on the safety and efficacy of these vaccines...". Again, no reference is provided, while the sentence states that at least some studies exist.

Response: Relevant references have been added. References 9 cover this.

4) In the "Introduction" section, line 51-52 of the original manuscript, authors say "...CKD is a significant risk factor [...] following infection with COVID-19. Among those infected with COVID-19...". As very clearly stated by the authors themselves, COVID-19 is the disease, not the virus. Thus, patients are infected by SARS-CoV-2, not by COVID-19. I would suggest authors change these and review other mentions of COVID-19, and change it by "SARS-CoV-2" when referring to the infection by the virus and not the disease.

Response: We have made a comprehensive review and revision of this confusion.

5) In the "Introduction" section, line 58, "Individuals with CKD can have previously benefitted...". "Can" should be removed.

Response: We fixed the error.

6) In the "Methods" section, "Data collection" subsection, line 96-97, "...type of vaccine, comorbidities, etc. will be collected". Authors used the future tense in this sentence. Do they refer to the follow-up specifically, which has not been completed? If not, "were collected" should be used instead.

Response: We fixed the error.

7) In the "Methods" section, "Data collection" subsection, line 98, "3 months" is repeated twice. In the second case it should state "6 months".

Response: We fixed the error.

8) In the "Methods" section, line 120, a space is missing between "probe" and "(biotinylated...)".

Response: We added a space here.

9) In the "Methods" section, line 122, what does the "(1:33.3)" ratio refer to?

Response: We fixed the error.

10) The questionnaire used for assessment of adverse effects is not detailed in the manuscript, nor in the Supplementary material. The questionnaire should be added to evaluate possible bias.

Response: We have added this questionnaire in the supplementary material 5.

11) In the "Results" section, line 169, "Presents as value ..." should say "Presented as value...". The same is true for table 2, line 177.

Response: We fixed these errors.

12) In the "Results" section, line 210, "Nabs" should be "NAbs".

Response: We fixed the error.

13) In the "Results" section, line 257-258, authors state "the positivity rate of NAbs (69.57% vs 79.89%, p=0.2854) was significantly different...". By the definition established by the authors in the "Methods" section, a p value of 0.2854 should not be significant.  Is the stated p value correct?

Response: We fixed the error. “was significantly different..." was revised as “was not significantly”.

14) In the "Discussion" section, lines 294-295, "...but this sound caution on this result" should be corrected.

Response: We deleted this sentence.

15) In the "Discussion" section, line 303, "The RECOVAC immune-responde atudy..." should be corrected to "study".

Response: We fixed the error.

16) In the "Discussion" section, when referencing work by other groups, authors mention their first name, or first name and last name. For example, "ChungYi et al." (line 301), "Borja Quiroga et al." (line 308), "Zeli Zhang et al." (line 310), among others. Traditionally, when citing works by other authors in text, usually only the surname is used. For example "Quiroga et al.".

Response: We fixed these incorrect citing.

17) In the "Discussion" section, line 316, the comma after "MBCs" should be removed. In the same line, a space should be added between "exposed" and "cells".

Response: We fixed these errors.

18) In the "Discussion" section, line 359-361, "The CKD patients showed good tolerance... two individuals presented with recurrent proteinuria, one of whom progressed to nephrotic syndrome". Taking into account that two of the CKD patients presented grade 3 or 4 adverse effects, stating that "The CKD patients showed good tolerance" is problematic, as it implies that all of them showed good tolerance. Alternatively, I propose changing it to "Most of the CKD patients showed good tolerance". Similary, in line 380-381, "...vaccines were well tolerated and showed good responses in the CKD population" should be changed. For example: "...showed good responses in the majority of the CKD population".

Response: We have corrected this inaccuracy according to your suggestion.

19) Reference 5 is almost certainly incorrect. The title of the document is given as "Since January 2020 Elsevier Has Created a COVID-19 Resource Centre...", which looks like a generic message from Elsevier, probably automatically added to the first page of the referenced paper, and automatically but incorrectly detected as the title by a reference manager program. Authors should check that the references are correctly formatted.

Response: We fixed the reference here.